# Benzamide Derivatives Targeting the Cell Division Protein FtsZ: Modifications of the Linker and the Benzodioxane Scaffold and Their Effects on Antimicrobial Activity

**DOI:** 10.3390/antibiotics9040160

**Published:** 2020-04-04

**Authors:** Valentina Straniero, Lorenzo Suigo, Andrea Casiraghi, Victor Sebastián-Pérez, Martina Hrast, Carlo Zanotto, Irena Zdovc, Carlo De Giuli Morghen, Antonia Radaelli, Ermanno Valoti

**Affiliations:** 1Dipartimento di Scienze Farmaceutiche, Università degli Studi di Milano, Via Luigi Mangiagalli, 25, 20133 Milano, Italy; lorenzo.suigo@unimi.it (L.S.); andrea.casiraghi@unimi.it (A.C.); 2Centro de Investigaciones Biológicas (CSIC), Ramiro de Maeztu 9, 28040 Madrid, Spain; victorsebastianperez@gmail.com; 3Exscientia, The Schrödinger Building, Oxford Science Park, Oxford OX4 4GE, UK; 4Pharmacy Faculty, University of Ljubljana, Aškerčeva cesta, 7, 1000 Ljubljana, Slovenia; martina.hrast@ffa.uni-lj.si; 5Dipartimento di Biotecnologie Mediche e Medicina Traslazionale, Università degli Studi di Milano, Via Vanvitelli, 32, 20129 Milano, Italy; carlo.zanotto@unimi.it (C.Z.); antonia.radaelli@unimi.it (A.R.); 6Veterinary Faculty, University of Ljubljana, Gerbičeva, 60, 1000 Ljubljana, Slovenia; irena.zdovc@vf.uni-lj.si; 7Department of Chemical – Pharmaceutical and Biomolecular Technologies, Catholic University “Our Lady of Good Counsel”, Rr. Dritan Hoxha, 1025 Tirana, Albania; carlo.degiulimorghen@unimi.it

**Keywords:** cell division protein FtsZ, benzamide, 1,4-benzodioxane, 1,4-benzoxathiane, multi-drug resistant *Staphylococcus aureus*, *Escherichia coli* N43

## Abstract

Filamentous temperature-sensitive Z (FtsZ) is a prokaryotic protein with an essential role in the bacterial cell division process. It is widely conserved and expressed in both Gram-positive and Gram-negative strains. In the last decade, several research groups have pointed out molecules able to target FtsZ in *Staphylococcus aureus*, *Bacillus subtilis* and other Gram-positive strains, with sub-micromolar Minimum Inhibitory Concentrations (MICs). Conversely, no promising derivatives active on Gram-negatives have been found up to now. Here, we report our results on a class of benzamide compounds, which showed comparable inhibitory activities on both *S. aureus* and *Escherichia coli* FtsZ, even though they proved to be substrates of *E. coli* efflux pump AcrAB, thus affecting the antimicrobial activity. These surprising results confirmed how a single molecule can target both species while maintaining potent antimicrobial activity. A further computational study helped us decipher the structural features necessary for broad spectrum activity and assess the drug-like profile and the on-target activity of this family of compounds.

## 1. Introduction

Antimicrobial resistance is a major cause of concern for public health worldwide, with the ever-growing diffusion of multidrug resistant pathogens and the consequent narrowing down of therapeutic options for previously treatable infections. 

According to a recent report by the Centers for Disease Control and Prevention (CDC), the grim predictions of a “post-antibiotic era” have come true, with an estimated over 3 million infections and nearly 50,000 deaths related to drug resistance per year in the U.S. alone [1]. The rise of drug resistant strains is largely propelled by the constant selective pressure associated with extensive use of antimicrobials in human and veterinary medicine and with their diffusion as environmental pollutants. In parallel, the number of novel antibiotics approved by international drug agencies decreased steadily over the last three decades, with only two new classes (lipopeptides and oxazolidinones) developed and approved in the past 20 years [2]. Notably, both classes have selective efficacy against Gram-positive bacteria, while the most recently approved innovative class for the treatment of Gram-negative infections are quinolones, dating back to 1962. In 2016, the World Health Organization (WHO) assessed the clinical relevance of 20 selected drug-resistant strains and compiled a priority list in order to guide future research and development investments. Critical priority was attributed to Gram-negative carbapenem-resistant *Acinetobacter baumannii, Pseudomonas aeruginosa* and third-generation cephalosporin-resistant Enterobacteriaceae (including *Escherichia coli* and *Klebsiella pneumoniae*) followed by high priority Gram-positive vancomycin-resistant *Enterococcus faecium* and methicillin-resistant *Staphylococcus aureus* [3].

Relying on more recent data, the WHO list updates and confirms the previous classification of ESKAPE *(E. faecium, S. aureus, K. pneumoniae, A. baumannii, P. aeruginosa* and *Enterobacter* species) pathogens as high priority pathogens, based on the incidence of nosocomial drug-resistant infections [4]. There are four main molecular mechanisms of antimicrobial resistance: i) mutation in genes encoding the target site, causing a reduction in drug affinity; ii) expression of specific enzymes, enhancing drug metabolism; iii) mutations of drug transporters and reduced drug influx; and iv) (over)expression of efflux pumps, resulting in increased drug efflux. Multiple mechanisms can coexist in the same bacterial cell, often with additive effects, resulting in the inability of the drug to reach the target’s binding site at an adequate inhibitory concentration [5]. 

Developing molecules able to target some of these ESKAPE pathogens without developing antimicrobial resistance is one of the great goals of medicinal chemistry today. This aim needs to deal with the efflux pumps, which represent a severe issue. They collectively form a ubiquitous cell detoxification system with multiple types of substrates, including antibiotics [6]. 

There are five main groups of efflux pumps involved in drug resistance: the resistance-nodulation-division (RND) family, the major facilitator superfamily (MFS), the ATP-binding cassette (ABC) superfamily, the small multidrug resistance (SMR) family and the multidrug and toxic compound extrusion (MATE) family [7]. Only RND pumps are exclusively found in Gram-negative species; the others are widely distributed in both Gram-positive and Gram-negative bacteria [8]. The main RND efflux pump is AcrAB-TolC, which is formed by a transporter protein in the inner membrane (AcrB), an accessory protein in the periplasm (AcrA), and an outer membrane protein channel (TolC) [9]. 

Aiming to achieve broad-spectrum antibiotics while considering the efflux pump susceptibility, structurally different compounds were recently developed. These molecules (zantrins **Z2** and **Z3** [10], and benzamide derivatives **TXY436** (which is a prodrug of the known **PC190723** [11]) and benzodioxane-benzamides **I** and **II** [12]) are summarized in Table 1, together with some relevant details. They all target the essential bacterial cell division protein FtsZ, a protein widely distributed among Gram-positive and Gram-negative strains, with which they interact at two different binding sites (the GTP-binding site for **Z2** and **Z3**, and the interdomain site for the others, which are better explained below). 

FtsZ is a GTP-binding protein, the bacterial homologue of eukaryotic tubulin [13], and it is highly conserved in most prokaryotic organisms [14]. Its structure can conventionally be divided into five specific portions, each characterized by its own degree of conservation: the *N*-terminal domain, the globular core, the *C*-terminal linker, the *C*-terminal tail and, lastly, the *C*-terminal variable region. Among them, the globular core contains the GTP-binding site, which is one of the two main binding sites of all known FtsZ inhibitors [15]. Considering the cruciality of the GTP-binding site for the physiological activity of FtsZ, the globular core is highly conserved among bacteria (both Gram positive and Gram negative) and mycobacteria.

The second binding site of FtsZ inhibitors is the interdomain site, a specific cavity located in a long cleft between the *C*-terminal domain and the H7 helix of FtsZ. Since this portion of the protein is not involved in any fundamental physiological function, its degree of conservation among bacteria is lower than the GTP-binding site. 

FtsZ is one of the main actors of the bacterial cell division process [16]. It is indeed a crucial protein of the divisome, a complex formed by a wide variety of protein components. The polymerization of FtsZ forms the Z-ring, which splits the mother bacterial cell into two daughter cells through its constriction [17]. FtsZ inhibition provokes the entire blockage of the division system and, consequently, it results in a bacteriostatic or a bactericidal effect. FtsZ essentiality in the division system has made this protein a fascinating target in antibiotic research. 

While working on FtsZ inhibitors, the previously mentioned researchers considered the insights coming from previous studies [18,19,20], in which it was shown how the genetic inactivation of RND pumps resulted in increased Gram-negative antimicrobial activity for a variety of other drugs. This finding strongly supports the role of these efflux systems in both intrinsic and acquired resistance for these microorganisms. 

Consequently, they employed a similar strategy to FtsZ inhibitors [10,11,12], in order to evaluate a possible restoring of the Gram-negative antimicrobial potency. Indeed, FtsZ inhibitors are usually described as limited to Gram-positive species, although this strict classification may be misleading, as highlighted in a recent review [15], since several players seem to contribute to the antimicrobial activity on different Gram strains. The most significant are here reported: (i) the FtsZ binding sites across different species seem to be slightly to strongly different, independently of the Gram staining; (ii) there is an evident imbalance in the number and in the variety of Gram-positive strains vs. Gram-negative strains tested, with a clear abundance of the former, specifically *Staphylococci*; (iii) the variety of Gram-negative strains tested was usually limited to *E. coli*, *K. pneumoniae* (both expressing AcrAB) and *P. aeruginosa*, which have an extensive array of chromosomally encoded RND efflux pumps, very limited membrane permeability, and a known intrinsic non-susceptibility to several antimicrobials.

Among the derivatives in Table 1, Compounds **I** and **II** resulted from our recent studies, in which we modified the substituent in position 7 of the 1,4-benzodioxane moiety. **I** and **II** showed MICs vs. Methicillin Resistant *S. aureus* (MRSA) and clinical isolates of multi-drug resistant *S. aureus* (*Sa*) between 0.6 and 2.5 µg/mL. Furthermore, morphometric analysis by transmission electron microscopy (TEM) on *Sa* cells incubated with them showed significant elongation in cellular dimension, bacterial swelling and cell desegregation, which are the typical alterations of cell division inhibition. 

**I** and **II** were developed after a precedent deep Structure Activity Relationship (SAR) study [21,22] and after the confirmation of FtsZ as the real target of our derivatives using two in vitro biochemical assays: a GTPase activity assay and a polymerization activity assay [23]. Specifically, we determined how our derivatives affect *Sa* FtsZ GTPase activity by kinetic measurement of inorganic phosphate release. An increase in polymerization was detected by SDS-PAGE, after the progressive addition of our compounds, and was quantified by the percentage FtsZ in the pellets using densitometric analysis [23].

In this work, with the aim of developing broad-spectrum antimicrobials and starting from *in-house* lead compound **III** (Figure 1), we developed novel derivatives, modifying the nature of the 1,4-benzodioxane ring or lengthening the distance between the 1,4-benzodioxane and the 2,6-difluorobenzamide moieties (Figure 2). 

Specifically, we substituted one or both oxygen atoms with sulfur, differently oxidized (**1**–**5**), as a conclusion of our study on the importance of the two oxygens of the 1,4-benzodioxane ring. We previously prepared and tested the chromanes **IV** and **V**, the tetrahydronapthalene **VI** [22], the benzoxazines **VII** and **VIII** and the tetrahydroquinoxaline **IX** [23]. None of these modifications resulted in a positive outcome for the antimicrobial activity. Nevertheless, our latest computational studies of our derivatives in the FtsZ binding site defined the cavity as characterized by narrowness and hydrophobicity [12]. As a result, we decided to include the sulfur as bioisostere of the oxygen atom, due to their similar chemical behavior and its stronger lipophilicity.

Moreover, in order to evaluate how deep is the binding site, we varied the length of the linker, moving from the methylenoxy chain of **III** to an ethylenoxy (**6** and **7**) or a propylenoxy one (**8**). As done before, we tested all the compounds against several Gram-positive *Sa* strains: methicillin sensitive, methicillin resistant and multi-drug resistant. Furthermore, we evaluate their activity not only vs. *E. coli* ATCC 25922 and extended-spectrum beta-lactamase-positive *E. coli*, but also on efflux-defective *E. coli* N43. Docking studies and predictions of a large number of physicochemical properties allowed us to better understand the antimicrobial profile of some compounds.

## 2. Results and Discussion

### 2.1. Design and Physico-chemical Profile Calculations

Compounds **1**–**8** were designed starting from the chemical structure of **III**, modifying the heteroatoms of the 1,4-benzodioxane ring for **1**–**6**, or lengthening the distance between the benzodioxane and the benzamide moieties for **7** and **8** (Figure 2). Considering the above-mentioned features of the FtsZ binding site and prior to the synthesis, we decided to calculate some important physico-chemical and drug-like properties *in silico*, as summarized in Table 2. 

The octanol/water partition coefficient (QlogPo/w) was chosen as a well-known parameter for the evaluation of their lipophilicity. Molecule dipole moment (Dipole) allowed us to assess the partial charge of the molecule and the impact of two potential charges, separated by a specific distance, which may affect molecular properties. Polar surface area (PSA) is a measure of the hydrogen-bonding capacity of the molecule and it summarizes the fractional contributions of all nitrogen and oxygen atoms to the surface area. Moreover, apparent Caco-2 cell permeability (QPPCaco) and apparent MDCK cell permeability (QPPMDCK) let us understand the permeability profiles, and thus the capability of reaching the target. 

As highlighted in Table 2, dipole and logP calculations were excellent for almost all the derivatives, being in the suitable range for drug-like molecules. Concerning permeability profiles, most of the molecules displayed a great permeability prediction; only sulfones **2** and **4** showed poor predicted values with high PSA, which may translate in poor cellular permeation. Overall, these four characteristics confirmed the high potential of our derivatives as drug candidates and moved us to the definition of peculiar synthetic pathways and to their consequent synthesis.

### 2.2. Chemistry

Scheme 1 shows the synthetic pathway of benzoxathiane derivatives **1** and **3**, together with the derived sulfones **2** and **4**. The benzoxathiane ring was achieved by treating the 1,2-mercaptophenol in a mixture of acetonitrile (ACN) and water (1/1) [24], in the presence of ethyl 2,3-dibromopropionate, prepared as reported in literature [25], with triethylamine (TEA) as a base. 

Operating with these reaction conditions, the relative percentages of 2- and 3-substituted benzoxathianes were 60% (**9**) and 40% (**10**), as expected. The ratio of regioisomers was easily defined through the comparison of ^1^H-NMR spectra, since the chemical shifts of the benzoxathiane protons were strongly dependent on the substitution position. 

Moreover, in order to maximize the general overall yield of the synthetic pathway, we decided to keep together **9** and **10** at this stage and to quantitatively separate the regioisomers only at the final step. Therefore, after their obtainment, the mixture of **9** and **10** sequentially underwent (1) reduction with LiAlH_4_ (**11** and **12**), (2) mesylation (**13** and **14**) and (3) *O*-alkylation of 2,6-difluoro-3-hydroxybenzamide with the mesylates **13** and **14**, with the final isolation and purification by flash chromatography, accomplishing pure **1** and **3**. As well as their precursors, **1** and **3**
^1^H-NMR spectra were significantly different and enabled the easy definition of the two chemical structures.

**2** and **4** were directly prepared through oxidation of **1** and **3**, using large equivalents of 3-chloroperoxybenzoic acid (*m*-CPBA) and operating at room temperature.

The synthesis of **5** (Scheme 2) started with the treatment of the commercial benzene-1,2-dithiol with ethyl 2,3-dibromopropionate in DMF and TEA as a base, affording **15** as quantitative product. The consecutive reduction with LiAlH_4_ (**16**) and mesylation yielded **17** that was condensed with the 2,6-difluoro-3-hydroxybenzamide (**5**).

The synthesis of **6** is reported in Scheme 3 and required the initial treatment of commercial 3-butenoic acid with bromine (**18**) and the consequent esterification of the carboxylic group (**19**). 1,2-mercaptophenol was then treated with **19** and TEA, in a mixture of ACN/H_2_O (1/1), accomplishing the 2-substituted-benzoxathiane **20** as the unique regioisomer. The benzoxathiine **20**, similarly to **9**, **10** and **15**, underwent reduction with LiAlH_4_ (**21**) and the tandem mesylation (**22**) and condensation with the 2,6-difluoro-3-hydroxy-benzamide, obtaining **6**.

Intermediate **19** was also used to obtain **7** (see Scheme 4). The pathway was very similar to the one described for **6**, but we used catechol instead of 1,2-mercaptophenol, accomplishing benzodioxane **23**, which underwent reduction (**24**), mesylation (**25**) and condensation with 2,6-difluoro-3-hydroxy-benzamide to afford **7**. 

Our initial idea to obtain **8** was the application of the synthetic pathway of **6**, while starting from commercial 4-pentenoic acid and catechol. Unfortunately, after the initial quantitative bromination and esterification, during the condensation with catechol we obtained, isolated and characterized methyl 4-bromopent-4-enoate as main reaction by-product (Figure 3). Literature data [26] were in line with our ^1^H-NMR spectra. Different reaction variables were evaluated and changed, but the reaction outcome was always the same. 

We therefore moved to a different synthetic pathway, in which we first include the 2,6-difluorobenzamide moiety and we then suitably functionalized the double bond. As reported in Scheme 5, we started from the mesylation of the commercial 4-penten-1-ol (**26**) and we soon condensed **26** with the 2,6-difluoro-3-hydroxy-benzamide (**27**). The alkene was then converted into the epoxide **28**, which was consequently reacted with the catechate, accomplishing **29** without any additional side-products. The final Mitsunobu reaction let the obtainment of the 1,4-benzodioxane ring and the final product **8**.

### 2.3. Antimicrobial Activity

Antimicrobial activity of **1–8** was evaluated on both Gram-positive (Table 3) and Gram-negative (Table 4) species. As a continuation of our previous studies, considering the Gram-positive *S. aureus*, we chose a methicillin-sensitive *S. aureus* (MSSA, ATCC 29213), a methicillin-resistant *S. aureus* (MRSA, ATCC 43300) and two clinical isolates of *S. aureus* showing multi-drug resistance (MDRSA). MDRSA 12.1 shows resistance towards gentamicin, kanamycin, rifampicin, streptomycin, sulfamethoxazole and tetracycline, while MDRSA 11.7 is resistant to ciprofloxacin, clindamycin, erythromycin, quinupristin and dalfopristin in association, tetracycline, tiamulin and trimethoprim.

The inhibitory ability of **1–8** was evaluated by determining the MIC, i.e., the lowest compound dose (µg/mL) at which growth is inhibited, as well as the minimal bactericidal concentration (MBC), i.e., the minimal dose (µg/mL) at which cell growth is irreversibly blocked after compound removal. The compounds showing the most promising activities vs. MRSA were also assessed for their cytotoxicity on human MRC-5 cells. TD90, the concentration (µg/mL) able to reduce by 90% the viability of these cells, was determined together with the therapeutic index (TI), expressed as the ratio between TD90 and MBC values. 

Considering all the results presented in Table 3, different conclusions can be made for compounds **1–5**, presenting a modified 1,4-benzodioxane scaffold and **6–8**, having an elongated linker between the heteroaromatic moiety and the 2,6-difluorobenzamide.

Starting from the former (**1–5**), the outstanding antimicrobial potency of **1** emerges. This surprising behavior confirm two important features: i) the maintenance of the oxygen in position 1 is essential for potent antibacterial activity on *S. aureus*; ii) the lipophilicity of this scaffold should be sufficiently high to properly interact with FtsZ. Indeed, the increased lipophilicity of **1,** having a sulfur atom in position 4 instead of the oxygen of **III**, results in five-fold higher MICs on MSSA and MRSA. The inhibitory activity of **1** on MDRSA has a similar outcome; despite its slightly weakened potency, it is four- and eight-fold stronger than **III** on MDRSA 12.1 and 11.7, respectively.

The antimicrobial profile of **3** emphasizes the importance of maintaining the oxygen atom in position 1. The retention of a comparable lipophilicity, while inverting the position of the two heteroatoms, results in a reduced interaction with the target and a consequent mild antimicrobial activity.

The MICs ratio value between **1** and **3** is 20, which is entirely in line with what was previously observed for chromanes **IV** (MIC vs. MSSA: 5 µg/mL) and **V** (MIC vs. MSSA: 100 µg/mL) [22]. This trend corroborates the essentiality of oxygen (1) for a strong interaction with FtsZ. A similar trend could be detected comparing the MICs of four derivatives with progressively more lipophilic heteroatoms in position 4: **VII**, **IV, III** and **1**. Indeed, moving from the benzoxazine **VII** to the chromane **IV**, to the benzodioxane **III** and lastly to the benzoxathiane **1,** we observe a growing antibacterial potency.

Moreover, the position of the sulfur atom affects also the human cytotoxicity, as shown by the results in Table 3. Indeed, derivative **1** shows an impressive and promising TI, strongly superior to **3** and **5**. Compound **5** shows an antimicrobial profile that is similar to **III**. Such a result could derive from the balance between the higher lipophilicity due to the presence of two sulfur atoms, and the absence of oxygen (1). Therefore, these two main determinants seem to equally contribute to the interaction with FtsZ and the consequent antibacterial activity. 

Lastly, in sulfones **2** and **4** the oxidation of the sulfur atoms brought to a crucial loss in lipophilicity and consequently in antimicrobial activity, further corroborating our hypothesis of a hydrophobic binding pocket.

Compounds **6–8** were designed starting from our previous computational studies [12], in order to assess the depth of the binding cavity. As highlighted from the MICs in Table 3, **6** and **7**, bringing the ethylenoxy chain, retain good ability to interact with FtsZ. Conversely, the propylenoxy bridge of **8** seems to be not well tolerated and does not allow a proper fitting of the compound. 

Finally, the MBC/MIC ratios for the whole series of compounds is quite surprising. The initial value of 16 for **III** significantly decreases to 1 or 2 for the most promising derivatives (**1**, **3**, **5**–**7**), enhancing their bactericidal effect.

Furthermore, we considered our recent and promising results of **I** and **II** [12], which highlights their capability to penetrate the *E. coli* membrane and to have an antibacterial activity on this bacterium. We previously shown how the genetic inactivation of RND-type efflux pump AcrAB (in *E. coli* N43) imparted a strong potency of **I** and **II** on this pathogen, confirming they are substrates of this efflux pump.

We thus tested **1–8** and **III** initially on the reference sensitive *E. coli* ATCC 25922 and on the extended-spectrum of beta-lactamase-positive (ESBL) *E. coli*, as Gram-negative bacteria. Moreover, in order to evaluate the susceptibility of this series of compounds to the efflux pump AcrAB, we employed also the above-mentioned *E. coli* N43, which is the knockout of the AcrAB cell membrane pump. 

The values in Table 4 show encouraging antimicrobial activities, especially for compound **7**. Its MIC is half the ones of the in-house reference compounds **I** and **II**. Moreover, surprisingly **7** has MICs vs. *S. aureus* and *E. coli* N43 of the same order of magnitude, confirming FtsZ as a valid and real target for both Gram-positive and Gram-negative strains.

### 2.4. Microscopy Evaluation

After having assessed the antimicrobial profile of this class of derivatives, we chose **1** to be further analyzed by transmission electron microscopy (TEM), in order to evaluate any morphological changes in septum formation, as done before [12,22,23] as a simple FtsZ target validation assay. We cultured *S. aureus* ATCC 43300 alone or in the presence of **1** and 2,6-difluorononyloxybenzamide (**DFNB**) [27] as a reference FtsZ inhibitor (positive control). No morphological changes were noticed in untreated *S. aureus*, as reported in Figure 4. On the contrary, when treated with **1** or **DFNB**, *S. aureus* cells showed strong alterations in the septum formation and bacterial desegregation, with consequent loss of the cytoplasmic content. These are the typical effects of the inhibition of FtsZ assembly and consequently of the block of bacterial cell division. 

### 2.5. Computational Studies

Physical-chemical properties predictions allowed us to design and propose compounds with a promising drug-like profile. Moreover, these properties gave preliminary insights into the structure–activity relationship of some compounds of this family. Indeed, they might explain the lack of activity of both compounds **2** and **4** in the MRSA and MSSA biological assays, because of their poor permeability profiles. On the contrary, the rest of the compounds displayed excellent predicted permeation profiles, being able to exert their action on FtsZ inside the bacteria. 

We decided to further evaluate derivatives **1**–**8** from a computational perspective; docking analyses were performed to study the binding mode of these molecules in the interdomain active site of the protein. According to our previous docking model, three hydrogen bonds drive the interaction between the 2,6-difluorobenzamide moiety and the *S. aureus* FtsZ protein [12]. Here, we confirmed that the amide group of the 2,6-difluorobenzamide scaffold can interact via two hydrogen bond donors between the NH_2_ and Val207 and Asn263 and via a hydrogen bond acceptor between the carbonyl and Leu209. Figure 5 depicts the docking results for the most active compound of this series, **1**. 

On the other hand, our previous studies highlighted the specific FtsZ region involved in binding the 1,4-benzodioxane. Indeed, this scaffold extends its conformation towards the inner part of the pocket and this hydrophobic subpocket is surrounded by hydrophobic residues such as Met98, Phe100, Val129, Ile162, Gly193, Ile197, Val214, Met218, Met226, Leu261 and Ile311 [12]. The high hydrophobicity of this subpocket suggests that apolar tails are preferred, while polar moieties are not well-tolerated. This finding, in combination with the poor permeation predictions, justifies the lack of activity of sulfones **2** and **4**, products of sulfur atom oxidation in the benzoxathiane scaffold. In addition, docking results of **2** and **4** clearly displayed significant clashes with FtsZ, as a result of the steric hindrance by the sulfonyl group.

Moreover, the second main aim of our computational study was to decipher the influence of the linker length in FtsZ interaction and thus in the antimicrobial activity. 

The biological data in Table 3 suggest that the presence of an ethylenoxy linker (compounds **6** and **7**) let the maintenance of a biological activity that is comparable to the one of their homologues **1** and **III**, respectively. We thus in silico analyzed **6** and **7** and we compared their binding poses. The docking results were almost the same, as exemplified in Figure 6 presenting derivative **7**. 

Finally, we considered the propylenoxy chain of **8**. Docking studies showed that several binding modes were possible, due to the flexibility of the linker. As a result, a rearrangement of the 2,6-difluorobenzamide moiety may be required for properly fitting the subpocket binding site. This might translate to the decrease in the biological activity of **8**, when compared to its homologues **III** and **7**, presenting the methylenoxy and ethylenoxy linkers, respectively.

In conclusion, all the results presented in this paper entirely support our hypothesis that the deep cavity of the FtsZ interdomain binding site, where the 1,4-benzodioxane scaffold fits, is characterized by narrowness and hydrophobicity. 

As a result, the transformation of the 1,4-benzodioxane moiety into a more lipophilic one, as the 2-substituted 1,4-benzoxathiane or the 2-substituted 1,4-benzodithiane, let a proper accommodation into this pocket. The sulfur atom, among all the heteroatoms evaluated in these years, shows to be beneficial for the substitution of 1,4-benzodioxane oxygen(4), obtaining potent antimicrobials with a bactericidal effect and a strong effect on bacteria morphological changes. 

Moreover, the promising antimicrobial activities on *E. coli* N43 suggest that this lipophilic cavity is almost conserved among Gram-negative and Gram-positive strains. Unfortunately, to our knowledge, there are no *E. coli* FtsZ crystallographic structures reported in literature thus far, including the interdomain binding site, where our class of derivatives binds. As soon as the X-ray structure will be available, we could move to a better design of novel broad spectrum FtsZ inhibitors, aided by sharp computational studies.

Lastly, we confirmed the susceptibility of our compounds to the *E. coli* AcrAB cell membrane pump, corroborating our preliminary results for benzodioxane-based inhibitors [12]. Similar findings were reported in a previous study by Kaul et al. [11], in which the antibacterial effect of TXA436 (Table 1) was restored in selected Gram-negative strains after the genetic inactivation of AcrAB or its inhibition by the polyselective RND efflux pump inhibitor phenylalanine arginyl β-naphthylamide (PaβN), reaching MIC of 8 μg/mL. Similarly, synergy studies with specific efflux pump inhibitors (EPIs) will be needed in order to better characterize the efflux pump susceptibility and the activity of benzodioxane-based FtsZ inhibitors in a larger number of Gram-negative strains, as well as to evaluate the potential of EPI-FtsZ inhibitors associations as broad spectrum therapeutic options for multidrug resistant bacterial infections. 

## 3. Materials and Methods 

### 3.1. Chemistry

All reagents (1,2-mercaptophenol, benzene-1,2-dithiol, catechol, LiAH_4,_ mesyl chloride, K_2_CO_3,_ TEA, *m*-CPBA, trimethyl orthoformate, H_2_SO_4_ conc) were purchased from Sigma Aldrich and were used without further purification. The solvents, such as ACN, THF, DCM, DMF, methanol and acetone, were purchased from Sigma Aldrich. 

Silica gel matrix, with fluorescent indicator 254 nm, was used in analytical thin-layer chromatography (TLC on aluminium foils), and silica gel (particle size 40-63 µm, Merck) was used in flash chromatography on Biotage IsoleraTM One or Sepachrom Puriflash XS 420; visualizations were accomplished with UV light (λ 254 nm). 

^1^H-NMR spectra were measured by Varian Mercury 300 NMR spectrometer/Oxford Narrow Bore superconducting magnet operating at 300 MHz. ^13^C-NMR spectra were acquired operating at 75 MHz. Chemical shifts (δ) are reported in ppm relative to residual solvent as internal standard. Signal multiplicity is used according to the following abbreviations: s = singlet, d = doublet, dd = doublet of doublets, t = triplet, q = quadruplet, m = multiplet, bs = broad singlet. The final products, **1–8,** were analyzed by reverse-phase HPLC using a Waters XBridge C-18 column (5 μm, 4.6 mm × 150 mm) on an Elite LaChrom HPLC system with a diode array detector. Mobile phase: A, H_2_O with 0.10% TFA; B, acetonitrile with 0.10% TFA; gradient, 90% A to 10% A in 25 min with 35 min run time and a flow rate of 1 mL/min. Their purity was quantified at peculiar λ max, depending on each sample and resulted to be >95%. The relative retention times are reported in each experimental section. Melting points were determined by DSC analysis over a TA Instruments DSC 1020 apparatus. 

#### Synthesis 

**Ethyl 1,4-benzoxathian-2-carboxylate (9)** and **ethyl 1,4-benzoxathian-3-carboxylate (10):** TEA (0.50 mL, 3.52 mmol) was added dropwise to a solution of ethyl-2,3-dibromopropionate (0.44 g, 1.76 mmol) into a mixture 1/1 of ACN and water (10 mL). After stirring at room temperature for 30 min, 1,2-mercaptophenol (0.22 g, 1.76 mmol) was slowly added. The reaction mixture was stirred at RT for 72 h, and then, it was diluted with diethyl ether and water. The organic phase was dried over Na_2_SO_4_, filtered and concentrated to give a yellow oil as the residue. The crude was purified by flash chromatography on silica gel, using cyclohexane/ethyl acetate 95/5 as the elution solvent; the yielded ratio between **9** and **10** was quantified by ^1^H NMR and resulted in 60% for **9** and of 40% for **10**. Yield: 65%

**Ethyl 1,4-benzoxathian-2-carboxylate (9). ^1^H NMR (300 MHz, CDCl_3_, δ):** 7.04 (m, 2H), 6.96 (d, *J* = 7.7 Hz, 1H), 6.87 (t, *J* = 7.4 Hz, 1H), 4.99 (dd, *J* = 5.3, 3.8 Hz, 1H), 4.27 (q, *J* = 7.1 Hz, 2H), 3.29 (d, *J* = 3.8 Hz, 1H), 3.28 (d, *J* = 5.3 Hz, 1H), 1.28 ppm (t, *J* = 7.1 Hz, 3H). 

**Ethyl 1,4-benzoxathian-3-carboxylate (10). ^1^H NMR (300 MHz, CDCl_3_, δ):** 7.06 (m, 1H), 7.00 (dd, *J* = 7.6, 2.0 Hz, 1H), 6.88 (t, *J* = 7.4 Hz, 2H), 4.56 (dd, *J* = 11.5, 3.0 Hz, 1H), 4.46 (dd, *J* = 11.5, 6.5 Hz, 1H), 4.25 (q, *J* = 7.1 Hz, 2H), 4.12 (dd, *J* = 6.5, 3.0 Hz, 1H), 1.30 ppm (t, *J* = 7.1 Hz, 3H). 

**2-hydroxymethyl-1,4-benzoxathiane (11)** and **3-hydroxymethyl-1,4-benzoxathiane (12):** LiAlH_4_ (85 mg, 2.23 mmol) was suspended in dry THF (5 mL) at 0 °C under nitrogen atmosphere. The mixture of **9** and **10** (0.5 g, 2.23 mmol) was dissolved in THF (5 mL) and was added to the reaction. The mixture was warmed to RT and stirred for 30 minutes; at completion, it was cooled to 0 °C and slowly quenched with ethyl Acetate (5 mL). Further ethyl acetate (10 mL) was added, the organic layer was washed with brine (3 × 10 mL), dried over Na_2_SO_4_ and concentrated under vacuum to give 0.38 g (81%) as a mixture of 60% **11** and 40% **12** as a colorless oil. 

**2-hydroxymethyl-1,4-benzoxathiane (11). ^1^H NMR (300 MHz, CDCl_3_, δ):** 7.03 (m, 2H), 6.86 (m, 2H), 4.32 (m, 1H), 3.90 (dd, *J* = 11.7, 4.2 Hz, 1H), 3.84 (dd, *J* = 11.7, 5.6 Hz, 1H), 3.14 (dd, *J* = 13.0, 9.0 Hz, 1H), 2.97 ppm (dd, *J* = 13.0, 2.1 Hz, 1H). 

**3-hydroxymethyl-1,4-benzoxathiane (12). ^1^H NMR (300 MHz, CDCl_3_, δ):** 7.02 (m, 2H), 6.86 (m, 2H), 4.50 (dd, *J* = 11.7, 4.2 Hz, 1H), 4.34 (dd, *J* = 11.7, 2.1 Hz, 1H), 3.88 (m, 2H), 3.42 ppm (ddd, *J* = 7.2, 4.2, 2.1 Hz, 1H).

**2-mesyloxymethyl-1,4-benzoxathiane (13) and 3-mesyloxymethyl-1,4-benzoxathiane (14):** Mesyl chloride (0.55 mL, 7.13 mmol) was added dropwise to a solution of **11** and **12** (1.0 g, 5.49 mmol) and TEA (1.0 mL, 7.13 mmol) in DCM (5 mL) at 0 °C. The mixture was stirred at room temperature for 3 h, diluted with DCM (20 mL), washed firstly with 10% aqueous NaHCO_3_ (10 mL), secondly with 10% aqueous HCl (10 mL) and finally with brine (10 mL), dried over Na_2_SO_4_, filtered and concentrated under vacuum to yield 1.58 g (80%) of **13** and **14** as an oil residue. 

**2-mesyloxymethyl-1,4-benzoxathiane (13). ^1^H NMR (300 MHz, CDCl_3_, δ):** 7.04 (m, 2H), 6.88 (ddd, *J* = 14.2, 7.7, 1.2 Hz, 2H), 4.57 (m, 1H), 4.45 (m, 2H), 3.10 (s, 3H), 3.20–2.99 ppm (m, 2H).

**3-mesyloxymethyl-1,4-benzoxathiane (14). ^1^H NMR (300 MHz, CDCl_3_, δ):** 7.03 (m, 2H), 6.89 (m, 2H), 4.63 (ddd, *J* = 11.8, 3.2, 1.5 Hz, 1H), 4.42 (m, 2H), 4.24 (m, 1H), 3.59 (dddd, *J* = 9.5, 6.4, 3.2, 1.7 Hz, 1H), 3.07 ppm (s, 3H).

**3-(1,4-benzoxathiane-2-yl)-2,6-difluorobenzamide (1) and 3-(1,4-benzoxathiane-3-yl)-2,6-difluorobenzamide (3):** Potassium carbonate (0.42 g, 3.01 mmol) was added to a solution of 2,6-difluoro-3-hydroxybenzamide (0.50 g, 2.88 mmol) in dry DMF (2 mL). After stirring at room temperature for 30 min, a solution of **13** and **14** (0.71 g, 2.74 mmol) in DMF (2 mL) was added. The reaction mixture was stirred at 60 °C for 16 h, concentrated under vacuum, diluted with Ethyl Acetate (20 mL), washed with brine (3 × 10 mL), dried over Na_2_SO_4_, filtered and concentrated, to give a residue, which was purified by flash chromatography on silica gel. Elution with 9/1 Cyclohexane/Ethyl Acetate gave **1** (35%) and **3** (20%) as white solids. mp (**1**): 117 °C and mp (**3**): 109–112 °C.

**3-(1,4-benzoxathiane-2-yl)-2,6-difluorobenzamide (1).** Tr (HPLC): 13.3 min (A% = 98.0% at λ = 288 nm). **^1^H NMR (300 MHz, d_6_-DMSO, δ):** 8.14 (bs, 1H), 7.85 (bs, 1H), 7.29 (m, 1H), 7.04 (m, 3H), 6.85 (m, 2H), 4.56 (m, 1H), 4.31 (m, 2H), 3.29 (dd, *J* = 14.1, 13.1 Hz, 1H), 3.15 ppm (dd, *J* = 13.1, 8.4 Hz, 1H). ^**13**^**C NMR (75 MHz, d_6_-DMSO, δ):** 161.7, 152.6 (dd, *J* = 240.5, 6.9 Hz), 151.2, 148.4 (dd, *J* = 247.4, 9.1 Hz), 143.2 (dd, *J* = 10.9, 2.8 Hz), 127.7, 126.2, 122.1, 118.8, 117.5, 117.1 (dd, *J* = 24.0, 20.6 Hz), 116.5 (d, *J* = 9.1 Hz), 111.5 (dd, *J* = 22.9, 3.4 Hz), 73.1, 71.2, 26.1 ppm. 

**3-(1,4-benzoxathiane-3-yl)-2,6-difluorobenzamide (3):** Tr (HPLC): 13.5 min (A% = 95.1% at λ = 288 nm). **^1^H NMR (300 MHz, d_6_-DMSO, δ):** 8.13 (bs, 1H), 7.85 (bs, 1H), 7.27 (m, 1H), 7.03 (m, 3H), 6.86 (m, 2H), 4.48 (dd, *J* = 11.8, 4.3 Hz, 1H), 4.24 (m, 3H), 3.88 ppm (m, 1H). ^**13**^**C NMR (75 MHz, d_6_-DMSO, δ):** 161.7, 152.6 (dd, *J* = 240.5, 6.8 Hz), 151.5, 148.3 (dd, *J* = 247.3, 8.0 Hz), 143.0 (dd, *J* = 10.9, 2.8 Hz), 127.7, 126.0, 122.5, 118.7, 117.3, 117.1 (dd, *J* = 24.6, 20.0 Hz), 116.4 (d, *J* = 9.2 Hz), 111.5 (dd, *J* = 22.9, 4.6 Hz), 69.3, 65.3, 37.4 ppm. 

**3-(1,4-benzoxathiane-4,4-dioxide-2-yl)-2,6-difluorobenzamide (2):** m-CPBA (0.10 g, 0.59 mmol) was added to a cooled solution of **1** (0.10 g, 0.30 mmol) in acetone (5 mL) at 0 °C. The reaction mixture was warmed to RT and stirred for 2 h, then 10 mL of 10% aqueous NaHCO_3_ was added, followed by 15 mL of ethyl acetate. The organic phase was dried over Na_2_SO_4_, filtered, and concentrated to give an oily residue. The further treatment with DCM lets the precipitation of 0.03 g (28%) of **2** as a white solid. mp: 192 °C. Tr (HPLC): 10.1 min (A% = 95.9% at λ = 280 nm). **^1^H NMR (300 MHz, d_6_-DMSO, δ):** 8.13 (bs, 1H), 7.85 (bs, 1H), 7.76 (dd, *J* = 8.0, 1.6 Hz, 1H), 7.57 (m, 1H), 7.31 (td, *J* = 9.3, 5.3 Hz, 1H), 7.20 (m, 1H), 7.14-7.06 (m, 2H), 5.10 (m, 1H), 4.49 (dd, *J* = 11.3, 3.2 Hz, 1H), 4.42 (dd, *J* = 11.3, 5.1 Hz, 1H), 4.03 (dd, *J* = 14.2, 1.6 Hz, 1H), 3.82 ppm (dd, *J* = 14.2, 12.0 Hz, 1H). **^13^C-NMR (75 MHz, d_6_-DMSO, δ):** 162.0, 153.1, 152.7 (dd, *J* = 240.8, 6.7 Hz), 148.5 (dd, *J* = 247.5, 9.0 Hz), 142.9 (dd, *J* = 10.9, 3.4 Hz), 135.4, 125.7, 123.9, 122.9, 119.2, 117.1 (dd, *J* = 7.9, 1.9 Hz), 116.8 (dd, *J* = 24.7, 20.2 Hz), 111.7 (dd, *J* = 23.3, 3.7 Hz), 75.1, 71.0, 50.2 ppm.

**3-(1,4-benzoxathiane-4,4-dioxide-3-yl)-2,6-difluorobenzamide (4):** m-CPBA (0.06 g, 0.36 mmol) was added to a cooled solution of **2** (60 mg, 0.18 mmol) in acetone (5 mL) at 0 °C. The reaction mixture was warmed to RT and stirred for 2 h, then 10 mL of 10% aqueous NaHCO_3_ was added, followed by 15 mL of ethyl acetate. The organic phase was dried over Na_2_SO_4_, filtered, and concentrated to give an oily residue. The further treatment with DCM lets the precipitation of 0.03 g (28%) of **4** as a white solid. mp: 184 °C. Tr (HPLC): 10.3 min (A% = 97.9% at λ = 281 nm). **^1^H NMR (300 MHz, d_6_-DMSO, δ):** 8.11 (bs, 1H), 7.82 (bs, 1H), 7.76 (dd, *J* = 8.0, 1.6 Hz, 1H), 7.56 (ddd, *J* = 8.9, 7.3, 1.6 Hz, 1H 1H), 7.31 (td, *J* = 9.3, 5.3 Hz, 1H), 7.20 (m, 1H), 7.12-6.99 (m, 2H), 4.87 (dd, *J* = 13.1, 2.0 Hz, 1H), 4.81 (dd, *J* = 13.1, 5.6 Hz, 1H), 4.53 (m, 1H), 4.44-4.27 ppm (m, 2H). **^13^C-NMR (75 MHz, d_6_-DMSO, δ):** 161.5, 154.0, 152.8 (dd, *J* = 242.2, 6.7 Hz), 148.4 (dd, *J* = 249.8, 8.6 Hz), 142.7 (dd, *J* = 11.4, 3.4 Hz), 135.0, 126.0, 124.3, 122.8, 119.1, 117.1 (dd, *J* = 24.9, 19.9 Hz), 116.7 (d, *J* = 10.4 Hz), 111.5 (dd, *J* = 23.0, 4.0 Hz), 67.1, 64.3, 57.0 ppm.

**Ethyl 1,4-benzodithian-2-carboxylate (15):** TEA (0.56 mL, 4.03 mmol) was added dropwise to a solution of ethyl-2,3-dibromopropionate (0.52 g, 2.01 mmol) in DMF (5 mL). After stirring at room temperature for 20 min; therefore, 1,2-benzenedithiol (0.26 g, 1.83 mmol) in 2 mL of DMF was slowly added. The reaction mixture was stirred at RT for 2 h, and then it was concentrated under vacuum and then diluted with 15 mL of ethyl acetate. The organic phase was washed with brine (3 × 10 mL), dried over Na_2_SO_4_, filtered, and concentrated to give 0.38 g (86%) of **15** as a yellow oil. **^1^H NMR (300 MHz, CDCl_3_, δ):** 7.26 (m, 2H), 7.06 (m, 2H), 4.30 (t, *J* = 6.5 Hz, 1H), 4.23 (q, *J* = 7.2 Hz, 2H), 3.32 (d, *J* = 6.5 Hz, 2H), 1.28 ppm (t, *J* = 7.2 Hz, 3H). 

**2-hydroxymethyl-1,4-benzodithiane (16**): LiAlH_4_ (60 mg, 1.58 mmol) was suspended in dry THF (2 mL) at 0 °C under nitrogen atmosphere. The solution of **15** (0.38 g, 1.58 mmol) in THF (5 mL) was slowly added to the reaction. The mixture was then warmed to RT and stirred for 1 h; at completion, it was cooled to 0 °C and slowly quenched with ethyl acetate (5 mL). Further ethyl acetate (10 mL) was added, the organic layer was washed with brine (3 × 10 mL), dried over Na_2_SO_4_ and concentrated under vacuum to give 0.26 g (84%) of **16** as a brown oil. **^1^H NMR (300 MHz, CDCl_3_, δ):** 7.20 (m, 2H), 7.02 (m, 2H), 3.90 (dd, *J* = 11.1, 7.3 Hz, 1H), 3.81 (dd, *J* = 11.1, 6.0 Hz, 1H), 3.70 (m, 1H), 3.27 (dd, *J* = 13.4, 3.7 Hz, 1H), 3.10 ppm (dd, *J* = 13.4, 6.4 Hz, 1H). 

**2-mesyloxymethyl-1,4-benzodithiane (17):** Mesyl chloride (0.14 mL, 1.76 mmol) was added dropwise to a solution of **16** (0.25 g, 1.26 mmol) and TEA (0.25 mL, 1.76 mmol) in DCM (4 mL) at 0 °C. The mixture was stirred at that temperature for 3 h, diluted with DCM (20 mL), washed firstly with 10% aqueous NaHCO_3_ (10 mL), secondly with 10% aqueous HCl (10 mL) and finally with brine (10 mL), dried over Na_2_SO_4_, filtered and concentrated under vacuum to yield 0.34 g (quantitative) of **17** as a pink oil. **^1^H NMR (300 MHz, CDCl_3_, δ):** 7.18 (ddd, *J* = 9.4, 4.8, 2.3 Hz, 2H), 7.04 (m, 2H), 4.57 (t, *J* = 10.0 Hz, 1H), 4.40 (dd, *J* = 10.3, 5.1 Hz, 1H), 3.91 (m, 1H), 3.29 (dd, *J* = 14.0, 3.5 Hz, 1H), 3.19 (dd, *J* = 14.0, 5.1 Hz, 1H), 3.07 ppm (s, 3H).

**3-(1,4-benzodithiane-2-yl)-2,6-difluorobenzamide (5):** Potassium carbonate (0.19 g, 1.35 mmol) was added to a solution of 2,6-difluoro-3-hydroxybenzamide (0.22 g, 1.29 mmol) in dry DMF (2 mL). After stirring at room temperature for 30 min, a solution of **17** (0.34 g, 1.23 mmol) in DMF (2 mL) was added. The reaction mixture was stirred at 60 °C for 16 h, concentrated under vacuum, diluted with Ethyl Acetate (15 mL), washed with brine (3 × 10 mL), dried over Na_2_SO_4_, filtered and concentrated, to give a residue, which was purified by flash chromatography on silica gel. Elution with 6/4 Cyclohexane/Ethyl Acetate and further crystallization from chloroform gave 0.22 g (50%) of **5** as a white solid. mp: 134 °C. Tr (HPLC): 14.3 min (A% = 98.8% at λ = 268 nm). **^1^H NMR (300 MHz, d_6_-DMSO, δ):** 8.14 (bs, 1H), 7.85 (bs, 1H), 7.26 (m, 1H), 7.21 (m, 2H), 7.05 (m, 3H), 4.35 (m, 1H), 4.21 (m, 1H), 4.09 (m, 1H), 3.30 (dd, *J* = 13.6, 3.0 Hz, 1H), 3.18 ppm (dd, *J* = 13.6, 6.1, 2.5 Hz, 1H). ^**13**^**C NMR (75 MHz, d_6_-DMSO, δ):** 161.6, 152.6 (dd, *J* = 240.4, 6.8 Hz), 148.4 (dd, *J* = 247.2, 8.3 Hz), 143.0 (dd, *J* = 10.8, 3.2 Hz), 131.5, 131.0, 129.5, 128.8, 126.4, 125.6, 117.1 (dd, *J* = 24.9, 20.4 Hz), 116.6 (dd, *J* = 9.3, 2.1 Hz), 111.5 (dd, *J* = 22.8, 3.9 Hz), 71.6, 41.6, 29.9 ppm. 

**3,4-dibromobutyric acid (18)**: Bromine (2.98 mL, 58.1 mmol) was added dropwise to a solution of 3-butenoic acid (5.0 g, 58.1 mmol) in DCM (50 mL) at 0 °C. Once added, the reaction mixture was warmed to RT and kept stirring for 16 h. The reaction mixture was then quenched with aqueous 10% sodium thiosulfate solution, dried over Na_2_SO_4_, filtered and concentrated under reduced pressure, affording 13.25 g (93%) of **18** as a yellowish oil. **^1^H NMR (300 MHz, CDCl_3_, δ):** 9.58 (bs, 1H); 4.48 (m, 1H), 3.92 (dd, *J* = 10.4, 4.3 Hz, 1H), 3.72 (t, *J* = 10.4 Hz, 1H), 3.44 (dd, *J* = 17.1, 3.8 Hz, 1H), 2.93 ppm (dd, *J* = 17.1, 9.2 Hz, 1H). 

**Methyl 3,4-dibromobutyrate (19):** 0.5 mL of H_2_SO_4_ conc. was added dropwise to a solution of **18** (13.25 g, 53.8 mmol) and trimethyl orthoformate (11.35 g, 107.0 mmol) in methanol (150 mL) at 0 °C. Once added, the reaction mixture was warmed to RT and then refluxed, keeping stirring, for 16 h. The reaction mixture was then concentrated under reduced pressure, resumed with ethyl acetate (150 mL), washed with aqueous 10% NaHCO_3_ (2 × 100 mL) and brine (100 mL), dried over Na_2_SO_4_, filtered and concentrated under vacuum, affording 10.9 g (78%) of **19** as a yellowish oil. **^1^H NMR (300 MHz, CDCl_3_, δ):** 4.50 (m, 1H), 3.91 (dd, *J* = 10.4, 4.3 Hz, 1H), 3.76 (s, 3H), 3.72 (dd, *J* = 10.2, 2.7 Hz, 1H), 3.34 (dd, *J* = 17.1, 3.8 Hz, 1H), 2.87 ppm (dd, *J* = 16.7, 9.2 Hz, 1H). 

**Methyl (1,4-benzoxathian-2-yl)-acetate (20):** Methyl 3,4-dibromobutirrate (1.50 g, 5.77 mmol) was added dropwise to a solution of 1,2-mercaptophenol (0.73 g, 5.77 mmol) and TEA (1.16 mL, 11.54 mmol) in acetonitrile/water 1:1 (15 mL) and stirred at room temperature for 18 h. The reaction mixture is then added with ethyl acetate (15 mL), washed firstly with 10% aqueous NaOH (15 mL) and lastly with brine (15 mL). The organic phase is then dried over NaSO_4_, filtered and concentrated under vacuum, affording 1.01 g (78%) of **20** as a yellowish oil. **^1^H NMR (300 MHz, CDCl3, δ):** δ 7.15–6.93 (m, 2H), 6.94–6.74 (m, 2H), 4.70 (m, 1H), 3.74 (s, 3H), 3.14 (dd, *J* = 13.0, 2.3 Hz, 1H), 3.04 (dd, *J* = 13.0, 7.6 Hz, 1H), 2.87 (dd, *J* = 15.8, 6.7 Hz, 1H), 2.74 ppm (dd, *J* = 15.8, 6.5 Hz, 1H).

**2-hydroxyethyl-1,4-benzoxathiane (21):** LiAlH_4_ (30 mg, 1.78 mmol) was suspended in dry THF (2 mL) at 0 °C under nitrogen atmosphere. The solution of **20** (0.18 g, 0.71 mmol) in THF (5 mL) was slowly added to the reaction. The mixture was then warmed to RT and stirred for 1 h; at completion, it was cooled to 0 °C and slowly quenched with ethyl acetate (5 mL). Further ethyl acetate (10 mL) was added, the organic layer was washed with brine (3 × 10 mL), dried over Na_2_SO_4_ and concentrated under vacuum to give 0.13 g (92%) of **21** as an orange oil. **^1^H NMR (300 MHz, CDCl_3_, δ):** δ 7.10–6.91 (m, 2H), 6.91–6.75 (m, 2H), 4.44 (m, 1H), 4.02–3.79 (m, 2H), 3.12–2.95 (m, 2H), 2.15–1.87 (m, 2H), 1.78 ppm (bs, 1H).

**2-mesyloxyethyl-1,4-benzoxathiane (22):** Mesyl chloride (0.24 mL, 3.12 mmol) was added dropwise to a solution of **21** (0.51 g, 2.59 mmol) and TEA (0.44 mL, 3.12 mmol) in DCM (5 mL) at 0 °C. The mixture was stirred at that temperature for 3 h, diluted with DCM (20 mL), washed firstly with 10% aqueous NaHCO_3_ (10 mL), secondly with 10% aqueous HCl (10 mL) and finally with brine (10 mL), dried over Na_2_SO_4_, filtered and concentrated under vacuum to yield 0.71 g (quantitative) of **22** as a yellowish oil. **^1^H NMR (300 MHz, CDCl_3_, δ):** δ 7.11–6.93 (m, 2H), 6.85 (m, 2H), 4.61–4.32 (m, 3H), 3.04–3.01 (m, 2H), 3.03 (s, 3H), 2.18 ppm (m, 2H).

**3-(1,4-benzoxathian-2-yl)ethoxy)-2,6-difluorobenzamide (6)**: Potassium carbonate (0.40 g, 2.86 mmol) was added to a solution of 2,6-difluoro-3-hydroxybenzamide (0.49 g, 2.73 mmol) in dry DMF (5 mL). After stirring at room temperature for 30 min, a solution of **22** (0.70 g, 2.55 mmol) in DMF (5 mL) was added. The reaction mixture was stirred at 60 °C for 16 h, concentrated under vacuum, diluted with ethyl acetate (30 mL), washed with brine (3 × 20 mL), dried over Na_2_SO_4_, filtered and concentrated, to give a residue, which was purified by flash chromatography on silica gel. Elution with 1/1 cyclohexane/ethyl acetate and further crystallization from IPA gave 0.36 g (39%) of **6** as a white solid. mp: 109.75 °C. Tr (HPLC): 14.1 min (A% = 99.4% at λ = 282 nm). **^1^H NMR (300 MHz, d_6_-DMSO, δ):** 8.09 (bs, 1H), 7.82 (bs, 1H), 7.26 (dt, *J* = 9.3, 5.3 Hz, 1H), 7.10–6.94 (m, 3H), 6.89–6.74 (m, 2H), 4.38 (tdd, *J* = 8.1, 5.2, 2.1 Hz, 1H), 4.29–4.22 (m, 2H), 3.27 (dd, *J* = 13.2, 2.1 Hz, 1H), 3.07 (dd, *J* = 13.2, 8.1 Hz, 1H), 2.27–2.05 ppm (m, 2H). **^13^C NMR (75 MHz, d_6_-DMSO, δ):** 161.8, 152.3 (dd, *J* = 241.2, 6.8 Hz), 151.3, 148.3 (dd, *J* = 248.4, 8.4 Hz), 143.3 (dd, *J* = 10.9, 3.3 Hz), 127.5, 126.0, 121.9, 118.8, 117.7, 117.0 (dd, *J* = 24.9, 20.5 Hz), 115.9 (dd, *J* = 9.4, 2.5 Hz), 111.4 (dd, *J* = 22.9, 4.0 Hz), 71.5, 66.0, 34.1, 29.1 ppm.

**Methyl 1,4-benzodioxane-2-yl-acetate (23):** Potassium carbonate (4.78 g, 34.00 mmol) was added to a solution of catechol (1.2 g, 11.00 mmol) in acetone (20 mL). After stirring at room temperature for 30 min, a solution of **19** (3.00 g, 11.00 mmol) in acetone (20 mL) was added. The reaction mixture was stirred at 60 °C for 16 h, concentrated under vacuum, diluted with Ethyl Acetate (50 mL), washed with brine (3 × 40 mL), dried over Na_2_SO_4_, filtered and concentrated, to give a residue, which was purified by flash chromatography on silica gel. Elution with 8/2 Cyclohexane/Ethyl Acetate gave 1.98 g (87%) of **23** as a colourless oil. **^1^H NMR (300 MHz, CDCl_3_, δ):** 6.86 (m, 4H); 4.63 (tt, *J* = 6.7, 3.3 Hz, 1H), 4.32 (dd, *J* = 11.3, 2.2 Hz, 1H), 4.00 (dd, *J* = 11.3, 6.8 Hz, 1H), 3.75 (s, 3H), 2.80 (dd, *J* = 16.1, 6.8 Hz, 1H), 2.65 ppm (dd, *J* = 16.1, 6.7 Hz, 1H). 

**2-hydroxyethyl-1,4-benzodioxane (24):** LiAlH_4_ (0.37 g, 10.00 mmol) was suspended in dry THF (2 mL) at 0 °C under nitrogen atmosphere. A solution of **23** (1.98 g, 9.5 mmol) in THF (5 mL) was added dropwise to the mixture; once added, the reaction was warmed to RT and stirred for 1 h; at completion, it was cooled to 0 °C and slowly quenched with ethyl acetate (10 mL). Further ethyl acetate (10 mL) was added, the organic layer was washed with brine (3 × 20 mL), dried over Na_2_SO_4_ and concentrated under vacuum to give 1.60 g (93%) of **24** as a colorless oil. **^1^H NMR (300 MHz, CDCl_3_, δ):** 6.87 (m, 4H), 4.38 (m, 1H), 4.28 (dd, *J* = 11.3, 2.2 Hz, 1H), 3.94 (m, 3H), 1.91 ppm (m, 2H). 

**2-mesiloxyethyl-1,4-benzodioxane (25):** Mesyl chloride (1.00 mL, 13.32 mmol) was added dropwise to a solution of **24** (1.60 g, 8.88 mmol) and TEA (1.80 mL, 13.32 mmol) in DCM (20 mL) at 0 °C. The mixture was warmed to RT and stirred at that temperature for 2 h, diluted with further DCM (20 mL), washed firstly with 10% aqueous NaHCO_3_ (30 mL), secondly with 10% aqueous HCl (30 mL) and finally with brine (30 mL), dried over Na_2_SO_4_, filtered and concentrated under vacuum to yield 2.10 g (91%) of **25** as yellowish oil. **^1^H NMR (300 MHz, CDCl_3_, δ):** 6.87 (m, 4H), 4.48 (ddd, *J* = 10.2, 7.0, 5.1 Hz, 2H), 4.34 (m, 1H), 4.27 (dd, *J* = 11.4, 2.3 Hz, 1H), 3.96 (dd, *J* = 11.4, 7.0 Hz, 1H), 3.05 (s, 3H), 2.09 ppm (m, 2H).

**3-(1,4-benzodioxane-2-yl)ethoxy)-2,6-difluorobenzamide (7):** Potassium carbonate (0.42 g, 3.01 mmol) was added to a solution of 2,6-difluoro-3-hydroxybenzamide (0.50 g, 2.88 mmol) in dry DMF (5 mL). After stirring at room temperature for 30 min, a solution of **25** (0.71 g, 2.75 mmol) in DMF (5 mL) was added. The reaction mixture was stirred at 60 °C for 16 h, concentrated under vacuum, diluted with ethyl acetate (30 mL), washed with brine (3 × 20 mL), dried over Na_2_SO_4_, filtered and concentrated, to give a residue which was crystallized from 3/2 cyclohexane/ethyl Acetate, accomplishing 0.50 g (54%) of **7** as a white solid. mp: 96 °C. Tr (HPLC): 13.2 min (A% = 98.0% at λ = 278 nm). **^1^H NMR (300 MHz, d_6_-DMSO, δ):** 8.12 (bs, 1H), 7.85 (bs, 1H), 7.27 (td, *J* = 9.4, 5.3 Hz, 1H), 7.07 (t, *J* = 8.9 Hz, 1H), 6.84 (m, 4H), 4.37 (dd, *J* = 16.8, 5.7 Hz, 2H), 4.24 (t, *J* = 6.1 Hz, 2H), 3.97 (dd, *J* = 11.2, 7.2 Hz, 1H), 2.08 ppm (dd, *J* = 11.0, 6.2 Hz, 2H). **^13^C NMR (75 MHz, d_6_-DMSO, δ):** 161.8, 152.3 (dd, *J* = 239.9, 6.8 Hz), 148.3 (dd, *J* = 246.9, 8.4 Hz), 143.4, 143.3 (d, *J* = 10.8 Hz), 143.2, 121.8, 121.7, 117.6, 117.3, 117.1 (dd, *J* = 25.6, 21.0 Hz), 115.9 (d, *J* = 7.1 Hz), 111.4 (dd, *J* = 22.8, 3.9 Hz), 70.5, 67.6, 66.8, 30.4 ppm.

**Pent-4-en-1-yl methanesulfonate (26):** Mesyl chloride (1.08 mL, 13.93 mmol) was added dropwise to a solution of 4-penten-1-ol (1.00 g, 11.61 mmol) and TEA (1.90 mL, 13.93 mmol) in DCM (10 mL) at −15 °C. The mixture was stirred at that temperature for 3 h, diluted with DCM (20 mL), washed firstly with 10% aqueous NaHCO_3_ (10 mL), secondly with 10% aqueous HCl (10 mL) and finally with brine (10 mL), dried over Na_2_SO_4_, filtered and concentrated under vacuum to yield 1.52 g (80%) of **26** as a colorless oil. **^1^H NMR (300 MHz, CDCl_3_, δ):** 5.78 (ddt, *J* = 16.9, 10.2, 6.7 Hz, 1H), 5.12–4.97 (m, 2H), 4.23 (t, *J* = 6.5 Hz, 2H), 3.00 (s, 3H), 2.25–2.12 (m, 2H), 1.94–1.79 ppm (m, 2H).

**2,6-difluoro-3-(pent-4-en-1-yloxy)benzamide (27):** Potassium carbonate (0.48 g, 3.48 mmol) was added to a solution of 2,6-difluoro-3-hydroxybenzamide (0.57 g, 3.32 mmol) in dry DMF (5 mL). After stirring at room temperature for 30 min, a solution of **26** (0.52 g, 3.17 mmol) in DMF (5 mL) was added. The reaction mixture was stirred at 80 °C for 16 h, concentrated under vacuum, diluted with ethyl acetate (30 mL), washed firstly with 10% aqueous NaHCO_3_ (10 mL), secondly with 10% aqueous HCl (10 mL) and finally with brine (10 mL), dried over Na_2_SO_4_, filtered and concentrated, to give 0.71 g (93%) of **27** as a brown oil. **^1^H NMR (300 MHz, CDCl_3_, δ):** 6.99 (td, *J* = 9.1, 5.2 Hz, 1H), 6.86 (td, *J* = 9.1, 1.9 Hz, 1H), 5.98 (bs, 2H), 5.83 (ddt, *J* = 16.9, 10.2, 7.0 Hz, 1H), 5.12–4.92 (m, 2H), 4.01 (t, *J* = 6.4 Hz, 2H), 2.24 (dd, *J* = 14.1, 7.0 Hz, 2H), 1.98–1.81 ppm (m, 2H).

**2,6-difluoro-3-(3-(oxiran-2-yl)propoxy)benzamide (28):***m*-CPBA (0.31 g, 1.78 mmol) was added to a solution of **27** (0.43 g, 1.78 mmol) in DCM (5 mL) at 0 °C. The reaction mixture was stirred at room temperature for 18 h, diluted with DCM, washed with 10% aqueous NaHCO_3_ (2 × 15 mL) dried over NaSO_4_, filtered and concentrated to give 0.42 g (91%) of **28** as a colorless oil **^1^H NMR (300 MHz, CDCl_3_, δ):** 7.00 (td, *J* = 9.1, 5.2 Hz, 1H), 6.86 (td, *J* = 9.1, 1.9 Hz, 1H), 6.16 (bs, 1H), 6.03 (bs, 1H), 4.19–3.94 (m, 2H), 3.07–2.89 (m, 1H), 2.78 (dd, *J* = 4.8, 4.1 Hz, 1H), 2.51 (dd, *J* = 4.8, 2.7 Hz, 1H), 2.03–1.74 (m, 2H), 1.74–1.43 ppm (m, 2H).

**2,6-difluoro-3-((4-hydroxy-5-(2-hydroxyphenoxy)pentyl)oxy)benzamide (29):** NaH (0.04 g, 1.63 mmol) was suspended in dry DMF (5 mL) at 0 °C under nitrogen atmosphere. A solution of catechol (0.18 g, 1.63 mmol) in DMF (5 mL) was added dropwise and, after stirring at room temperature for 30 minutes, a solution of **28** (0.42 g, 1.63 mmol) in DMF (5 mL) was then added dropwise. The reaction mixture is heated at 100 °C and stirred for 72 h. The mixture was then cooled to RT and slowly quenched with 1/1 ethyl acetate and 10% aqueous HCl (40 mL). The phases were separated and the organic one was washed with brine (10 mL), dried over NaSO_4_, filtered and concentrated to yield 0.46 g (77%) of **29** as a reddish oil. **^1^H NMR (300 MHz, d_6_-DMSO, δ):** 8.62 (bs, 1H), 8.08 (bs, 1H), 7.80 (s, 1H), 7.20 (td, *J* = 9.3, 5.3 Hz, 1H), 7.03 (td, *J* = 9.3, 1.9 Hz, 1H), 6.89 (d, *J* = 6.9 Hz, 1H), 6.79–6.60 (m, 3H), 5.00 (s, 1H), 4.11–3.94 (m, 2H), 3.94–3.78 (m, 2H), 3.72 (dd, *J* = 8.8, 6.3 Hz, 1H), 1.95–1.45 ppm (m, 4H).

**3-(1,4-benzodioxane-2-yl)propoxy)-2,6-difluorobenzamide (8):** Under nitrogen atmosphere, to a solution of triphenylphosphine (0.48 g, 1.84 mmol) in THF (2 mL), DIAD (0.36 mL, 1.84 mmol) and a solution of **28** (0.45 g, 1.23 mmol) in THF (1 mL) were added dropwise at 0 °C. The reaction mixture was stirred at reflux for 24 h, concentrated under vacuum, diluted with ethyl acetate (10 mL), washed firstly with brine (10 mL), secondly with 10% aqueous NaOH (10 mL) and lastly with brine (10 mL), dried over NaSO_4_, filtered and concentrated, to give a residue which was purified by flash chromatography on silica gel. Elution with 1/1 cyclohexane/ethyl acetate gave 0.05 g (11%) of **8** as a white wax. Tr (HPLC): 13.7 min (A% = 99.2% at λ = 279 nm). **^1^H NMR (300 MHz, d_6_-DMSO, δ):** 8.11 (bs, 1H), 7.83 (bs, 1H), 7.20 (td, *J* = 9.3, 5.3 Hz, 1H), 7.04 (td, *J* = 9.0, 1.8 Hz, 1H), 6.87–6.70 (m, 4H), 4.31 (dd, *J* = 11.4, 2.2 Hz, 1H), 4.25–4.14 (m, 1H), 4.08 (t, *J* = 6.3 Hz, 2H), 3.87 (dd, *J* = 11.4, 7.5 Hz, 1H), 2.00–1.79 (m, 2H), 1.79–1.59 ppm (m, 2H). **^13^C NMR (75 MHz, d_6_-DMSO, δ):** 161.81, 152.13 (dd, *J* = 240.8, 6.8 Hz), 148.29 (dd, *J* = 248.2, 8.5 Hz), 143.46, 143.50 (dd, *J* =10.6, 3.2 Hz), 143.45, 121.72, 121.51, 117.51, 117.23, 117.0 (dd, *J* = 23.0, 18.6 Hz), 115.83 (dd, *J* = 9.3, 2.5 Hz), 111.33 (dd, *J* = 22.8, 4.0 Hz), 72.80, 69.59, 67.67, 27.22, 24.68 ppm.

### 3.2. Cells

Normal human lung fibroblasts (MRC-5) were grown in Dulbecco’s modified Eagle’s medium (DMEM) supplemented with 10% calf serum, 100 U/mL penicillin and 100 mg/mL streptomycin in an incubator at 5% CO_2_ atmosphere and 37 °C. The Gram-positive *Staphylococcus aureus* (methicillin-sensitive, MSSA ATCC 29213, and methicillin-resistant, MRSA ATCC 43300) and the Gram-negative *Escherichia coli* (ESBL, extended-spectrum beta-lactamase-positive) bacterial cells were grown in Luria-Bertani broth (LB) at 37 °C under constant shaking at 300 rpm.

### 3.3. Antibacterial Activity

#### 3.3.1. MSSA, MRSA and ESBL Protocols

The antibacterial activity was tested on both a methicillin-sensitive and a methicillin-resistant *S. aureus* strain, and an ESBL *E. coli* clinical isolate. All the compounds were dissolved at the final concentration of 20 mg/mL in dimethyl sulfoxide (DMSO) and serially diluted in LB. Fresh cell cultures were used at 10^5^ cells/mL in a final volume of 2 mL. Each bacterial sample was grown with different compound concentrations that ranged from 100 to 0.1 μg/mL. After incubation at 37 °C for 16 h in aerobic culture tubes, the concentration of prokaryotic cells was determined by optical density measurement, at 600 nm (OD_600_) in a SmartSpecTM 3000 spectrophotometer (Bio-Rad, Oceanside, CA, USA) to determine the MIC. To determine the MBC, the bacteria were then washed three times with LB, centrifuged at 900× *g* for 10 min at 4 °C, and the pellet resuspended in fresh LB. After overnight incubation at 37 °C, the absence of growth was confirmed by OD measurement. The presence of antibacterial activity, at any concentration tested, was established for values of absorbance < 0.1 OD_600._ All tests were performed in quadruplicate and for each series of experiments, both positive (no compounds) and negative (no bacteria) controls were included.

#### 3.3.2. MDRSA, *E. coli* N43, *E. coli* D22 Protocols

Antibacterial activities of compounds were determined by the broth microdilution method against MDRSA 11.7, MDRSA 12.1 and *E. coli* N43 following the European Committee on Antimicrobial Susceptibility Testing (EUCAST) recommendations and Clinical and Laboratory Standards Institute (CLSI) guidelines. Strains MDRSA ST-11.7 and ST-12.1 were obtained through the interlaboratory control organized by the EU Reference Laboratory—Antimicrobial Resistance. Only data on their resistance to antibiotics are known, but other data are not available.

Bacterial suspension of specific bacterial strain equivalent to 0.5 McFarland turbidity standard was diluted with cation-adjusted Mueller Hinton broth with TES (Thermo Fisher Scientific), to obtain a final inoculum of 10^5^ CFU/mL. Compounds dissolved in DMSO and inoculum were mixed together and incubated for 20 h at 37 °C. After incubation, the minimal inhibitory concentration (MIC) values were determined by visual inspection as the lowest dilution of compounds showing no turbidity. Tetracycline was used as a positive control on every assay plate.

### 3.4. Thiazolyl Blue Tetrazolium Bromide (MTT) Cytotoxicity Assay

Compounds showing an antibacterial activity at a concentration lower than 10 μg/mL were serially diluted in DMEM and tested on MRC-5 cells by the MTT cytotoxicity assay (Sigma, St Louis, MO, USA). Cells (10^4^ cells/well) were tested in a 96-well plate, in quadruplicate, using serially two-fold-diluted concentrations of the compound in 100 μL DMEM medium. After a 24-h incubation, the compound was removed, and the cells were overlaid with 1 mg/mL MTT in 100 μL serum-free DMEM for 3 h at 37 °C. The MTT solution was then replaced with DMSO for 10 min, and the absorbance was measured at 570 nm. The percentage of cytotoxicity was calculated by the formula [100 − (sample OD/untreated cells OD) × 100]. The compound concentration reducing cell viability by 90% was defined as the TD90 cytotoxic dose. The “therapeutic index” (TI) was also determined and defined as the ratio between TD90 and the MBC values.

### 3.5. Transmission Electron Microscopy

*S. aureus* ATCC 43300 (10^9^ cells/mL) were cultured in the presence of **1** and **DFNB** (positive control), at the same concentrations used for optical microscopy and, after 16 h incubation at 37 °C, the cells were harvested and processed for transmission electron microscopy, as already described [28]. Untreated *S. aureus* was used as negative control. After centrifugation at 3100× *g* for 5 min at room temperature, pelleted bacteria were fixed in 2.5% glutaraldehyde (Polysciences, Warrington, PA) in 0.1 M Na cacodylate buffer, pH 7.4, for 1 h at 4 °C, rinsed twice and post-fixed in Na cacodylate-buffered 1% OsO_4_ for 1 h at 4 °C. The samples were dehydrated through a series of graded ethanol solutions and propylene oxide and embedded in Poly/Bed 812 resin mixture. Ultrathin sections were obtained using a Reichert-Jung ultramicrotome equipped with a diamond knife. Samples were then stained with water-saturated uranyl acetate and 0.4% lead citrate in 0.1 M NaOH. The specimens were viewed under a Philips CM10 electron microscope.

### 3.6. Computational Studies

Ligand preparation. The preparation of the compounds synthesized in this work was performed before carrying out further computational studies. The preparation, together with the two-dimensional-to-three-dimensional conversion, was performed using LigPrep [29], included in the Schrödinger software package. In the preparation, we generated different steps such as the addition of hydrogens, the calculation of the ionization state of the molecules at a specific pH or the generation of potential tautomers. Additionally, low-energy ring conformations were generated for every molecule, followed by a final energy minimization step using the OPLS-2005 force [30,31]. With the aim of mimicking cell conditions, physiological pH states were used to prepare the molecules, all of them were desalted and the compounds were minimized as default in the last step. 

Protein preparation. The FtsZ crystal structure of *S. aureus* (PDB code: 5XDT) was used as the protein structure for the computational studies. The protein was prepared for docking studies following a protocol described in our previous studies [12].

Ligand Characterization. The compounds previously prepared were analyzed using the Qikprop module in the Schrödinger software package. QikProp allowed us to calculate and predict a total number of 44 pharmaceutically relevant ADME or ADME Tox properties among others. These properties include both simple molecular descriptors and relevant computational predictions for drug discovery. Properties were considered to assess key parameters such as lipophilicity or cellular permeability. 

Docking studies. Docking studies were performed by employing the Glide module included in the Schrödinger software package [32]. Once the protocol was validated in our previous studies [12], the FtsZ structure from *S. aureus* and the compounds synthesized in this work were used as a starting point for the analysis. Docking studies were performed applying the same protocol in terms of conformational search and evaluation parameters for all the molecules, considering the center of the grid the center of the previously crystallized ligand in the catalytic pocket. In the grid generation, a scaling factor of 1.0 in van der Waals radius scaling and a partial charge cut-off of 0.25 were used. The xtra precision (XP) mode with no constraints was applied during the docking [33]. The ligand sampling was flexible, epik state penalties were added to the docking score and an energy window of 2.5 kcal/mol was utilized for ring sampling. In the energy minimization step, distance dependent dielectric constant was 4.0 with a maximum number of minimization steps of 100,000. In the clustering, poses were discarded as duplicates if both the RMS deviation was less than 0.5 Å and the maximum atomic displacement was less than 1.3 Å.

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
