# Peer review of "Benzamide Derivatives Targeting the Cell Division Protein FtsZ: Modifications of the Linker and the Benzodioxane Scaffold and Their Effects on Antimicrobial Activity"

_antibiotics, 2020, doi:10.3390/antibiotics9040160_

Round 1
Reviewer 1 Report
Dear Editor,
The ms entitled “Benzamide derivatives targeting the cell division protein FtsZ: modifications of the linker and the benzodioxane scaffold and their effects on antimicrobial activity” by Straniero et al. reports on the antibacterial activity of synthetic molecules deriving from previously characterized structures of the group: derivatives of the 1,4-benzodioxane ring. Compounds also have linker changes separating their benzodioxane and benzamide moieties; authors suggested that this change may interfere with the interaction of their compounds and the target FtsZ
Compounds were evaluated against Gram-positive and Gram-negative bacteria exhibiting documented antimicrobial resistance
Authors show that the oxygen at position 1 is necessary for the anti-S. aureus action (specially of compound 1), and in addition, the sulfur at 4 seemed to contribute to an increase in lipophilicity enabling interaction with FtsZ
TI of 1 was also interesting
However, no action was detected against Gram-negative E. coli, apart from an indication of activity against an efflux pump mutant.
As I can see in previous works from the group, they could well expand their biological part by doing permeability analyses of membranes with simple microscopic studies, GTPase-activity using purified FtsZ, polymerization analyses, etc.
However, they quickly jumped from MIC/MBCs evaluations to computational dockings, ending up with the conclusion that “all the results presented in this paper entirely support our hypothesis that the deep cavity of the FtsZ interdomain binding site, where the 1,4-benzodioxane scaffold fits, is characterized by narrowness and hydrophobicity”
To my view, this ms reports on a series of compounds that showed truly interesting and promising antibacterial activity, but they have only MICs…I do not see anything else to support a publication in such preliminary format.
Author Response
We appreciate your comments on the manuscript and on the results (MICs and TI) we obtained. Considering the lack of target validation experiments in this manuscript, we decided not to perform them before the first submission, as the compounds of this paper have an high structural similarity if compared to the ones previously evaluated and as comparable conclusions were achieved by other researchers, basing on identical data. (see Hu et al BMCL, 2017, 27, 1854-1858; Hu et al BMCL, 2017, 27, 958-962; Zhang et al Scientific Report, 2019, 9, 8319). Nevertheless, we took a few images by microscopy before the forced closure of our lab due to COVID-19, so we were able to assess cell division inhibition by evaluating morphometric changes of S. aureus, when exposed to 1. A similar consideration was done by Sangeeta and co-workers (Microbial Pathogenesis 2018, 124, 258-265).
Moreover, considering FtsZ literature, the activity on Gram-negative was scarcely achieved so far, therefore we are quite confident our MICs on E. coli N43 are really promising. We will soon (unfortunately we strictly depend on COVID-19) move to perform some synergy studies with AcrAB Efflux Pump Inhibitors, in order to set-up an association able to really counteract E. coli infections.
We decided to consider your criticisms implementing the manuscript, especially when speaking about E. coli results. We will definitely move to perform in vitro specific assays on these derivatives, as soon as we could.
Reviewer 2 Report
An interesting paper exploring a promising avenue for the development of novel antibacterial drugs that might be used to treat antibiotic-resistant bacteria. I only have a few minor editing suggestions.
250 essentially involved in the bacterial cell division process
=> with an essential role in the bacterial cell division process
28 not so promising derivatives active on Gram-negative were found so far.
=> no promising derivatives active on Gram-negative had been found up to now.
39 for worldwide public health,
=> for public health worldwide,
43 have eventually became true
=> have come true
59 as highest priority pathogens,
=> as high priority pathogens
67 one of the great medicinal chemistry goals today.
=> one of the great goals of medicinal chemistry today.
108 An evaluation by SDS-page in the increase in polymerization was detected
=> An increase in polymerization was detected through SDS-page
194 was used also for the obtainment of 7
=> was also used to obtain 7
201 Our initial idea for the obtainment of 8
=> Our initial idea to obtain 8
260 almost analogous
Analogous means similar. Do you mean “almost similar”? Please clarify.
Author Response
We would thank the reviewer for his positive comments on our work and for his punctual suggestions, which were all considered, changing the manuscript accordingly.
Reviewer 3 Report
The authors have presented a manuscript on benzamide derivatives that target FtsZ protein in bacteria thereby affecting bacterial cell division. The text needs to be better organized into distinct sections and points listed below need to be addressed.
- The introduction can be split into sections with separate sub headings to make it easier for readers to follow the text. For example, mechanism of action can be one of the sections which details how the inhibitors work and potential shortfalls of inhibitors including the efflux pumps
- The introduction should contain more about FtsZ and which regions are commonly inhibited and why. Other than efflux pumps, why is there difference in antimicrobial activity? This part should be elaborated as it is relevant to this study.
- Figure 1 is already published in literature. It is not necessary to include that as a figure here. It might be more suitable though to summarize other details discovered along with these inhibitors in a table or graphically. Extra details such as MICs or which region of FtsZ is inhibited should be included in the figure/table.
- The section about efflux pump inhibitors use along with FtsZ inhibitors can be elaborated in discussion (line 343). Are there existing studies with other FtsZ inhibitors that have tested this? What have they observed? What do the authors hope to achieve by doing this? These points need to be discussed.
Author Response
We would acknowledge the reviewer for his favorable comments on our work. We considered all his punctual criticisms and we decided to change the manuscript:
- Implementing the details on FtsZ, its regions and its binding sites, and their conservation among species. Moreover, we stressed the most important reasons causing such strong differences in MICs among the tested strains;
- Transforming Figure 1 into Table 1, and including the details you suggested, since we considered your criticism very useful for help readers into understanding the state of art of this topic;
- Avoiding the splitting of the introduction into subsections, in order to have a consistent and not too long introduction;
- Enhancing the final considerations on the possible synergy studies with Efflux Pump Inhibitors. In FtsZ literature there is a single paper (Reference 11) in which authors used a polyselective RND efflux pump inhibitor phenylalanine arginyl β-naphthylamide (PaβN), reaching MIC of 8 μg/mL. we would perform similar studies, hopefully using specific Efflux Pumps Inhibitors (EPIs), as soon as COVID-19 will let us re-enter into our labs, in order to set-up potential EPI-FtsZ inhibitors associations as broad spectrum therapeutic options for multidrug resistant bacterial infections.
Reviewer 4 Report
General comments:
Straniero et al studied the antimicrobial effect of Benzamide derivatives against bacterial cell division protein FtsZ. Based on their previous studies, the authors identified several chemically modified molecules with potent efficacy in inhibiting pathogen growth against both gram-positive MDRSA and gram-negative E. coli with efflux pump defect. Further examining the chemical and physical properties of these molecules, they found both Oxygen (1) and linker elongation can enhance molecular interaction with FtsZ. They further validate their findings by in silico simulation of molecule-protein interaction. This work provided molecular understanding in the interaction between Benzamide derivatives and bacterial FtsZ protein and demonstrated that these small molecules are promising drug candidates to be used as broader spectrum antibiotics.
Specific comments:
Line 89-90: How conserve are FtsZ protein across gram-positive bacteria like MDRSA and gram-negative bacteria like E. coli? Authors can describe here similarity in protein sequences of FtsZ between known gram-positive MDRSA and gram-negative E. coli to illustrate the highly conserved nature of FtsZ protein.
Line 224-226: Can authors provide more information about the two MDRSA strains (e.g. source of isolation, public availability)? I was not able to find relevant information from the material section.
The information may provide additional resources for the antibiotic research community to examine the strain variation in response to modified compounds. For example, if the two strains have already been genome sequenced, their genetic mutation in the FtsZ gene may cause the variation in protein structure that links to different drug response shown in Table 2, and provide new leads to additional molecular understanding and potential targets.
Line 271: Since MBC/MIC ratio was mentioned here, add a column for this ratio in Table 2 can help readers to visualize these numbers directly.
Line 277: Are there known drug inhibitor of the RND-type efflux pump AcrAB? If yes, it would be interesting to test a cocktail of AcrAB inhibitor and Compound 7 against both E. coli wild type pathogens.
Line 300: Which bacterial FtsZ protein sequence was used here? Do these differences in protein sequences between bacterial strains affect the result of protein interactions here?
Line 639,642: For both MIC and MBC assay, what threshold of OD600 did authors use to define the effect of bacterial growth inhibition?
Author Response
We would thank the reviewer for his overall positive comments on our work. We considered his criticisms on the introduction and on the methods and we implemented the manuscript, as follows:
- Introducing an MBC/MIC column in Table 3 (former Table 2), in order to help readers during the discussion;
- Detailing the FtsZ strains used for computational studies and its degree of conservation among species;
- Enhancing the final considerations on the possible synergy studies with Efflux Pump Inhibitors. There are no known Efflux Pumps Inhibitors (EPIs) in literature that are specific vs AcrAB, but, as soon as COVID-19 will let us re-enter into our labs, we will soon set-up and perform EPI-FtsZ inhibitors associations with some potential EPIs coming from the research work of our colleagues, in order to define broad spectrum therapeutics;
- Detailing the MDRSA strains in the Materials and Methods section; we did our best, considering these clinical isolates are not genomic characterized, and only data on their resistance to antibiotics is known;
- Introducing in Materials and Method section the OD threshold for MBC and MIC assays.
Moreover, we would add some information, considering the specific question of the reviewer about the FtsZ degree of conservation. Indeed, since most of antibiotics used in therapy do not interact with the cell division cycle, we do not expect a significative difference in the FtsZ sequence among the wt- and clinically isolated MDR-/mutated strains of a particular bacterium, such as E. coli or S. aureus.
Nonetheless, we do not exclude that, within the evolution process, some point mutations might have occurred from non-resistant strains to MDR-strains. Literature data and database uniprot.org indeed report how FtsZ sequence similarity among MSSA/MRSA/MDRSA strains is in the range between 90 and 100%. Also considering E. coli, the sequence similarity of FtsZ of several strains is reported to be in the same range (90-100%). Besides, we are quite confident that those kind of point mutations do not affect the bacteriostatic/bactericide action of FtsZ inhibitors.
Furthermore, we also considered the interspecies degree of conservation, by aligning E. coli FtsZ (strain K12) (UniProt ID: P0A9A6) and S. aureus FtsZ (strain N315) (UniProt ID: P99108) using BLAST (https://blast.ncbi.nlm.nih.gov). The alignment shows a 51% of sequence identity, but this percentage increases to 74% if we consider positives, that means dealing not only with identical residues but also with residues having identical physical-chemical properties. These data demonstrate the high overall degree of conservation and justify the similar activity on E. coli N43 and S. aureus FtsZ of some of the presented compounds.
Round 2
Reviewer 1 Report
what I pointed out in my first review was not changed. They said they cannot implement experiments because of the covid crisis. If this is acceptable for the journal is beyond my decision now. Since my comments were in a broader aspect of the work, I recommend checking if they at least amend what the other reviewers requested.
Author Response
Dear reviewer,
We did our best in performing the morphometric assays, in order to validate FtsZ as the target of our compounds.
We understand your general considerations and we will surely move to set-up and always perform further validation assays, prior to our future paper submissions.